# Modified Signaling of Membrane Formyl Peptide Receptors in NADPH-Oxidase Regulation in Obesity-Resistant Mice

**DOI:** 10.3390/membranes13030306

**Published:** 2023-03-06

**Authors:** Irina Tikhonova, Alsu Dyukina, Elvira Shaykhutdinova, Valentina Safronova

**Affiliations:** 1Institute of Cell Biophysics, Russian Academy of Sciences, Institutskaya St., 3, 142290 Pushchino, Russia; 2Branch of Shemyakin-Ovchinnikov Institute of Bioorganic Chemistry, Russian Academy of Sciences, Prospect Nauki, 6, 142290 Pushchino, Russia

**Keywords:** membrane receptors, NADPH-oxidase, MAP kinases, granulocyte, inflammation, high-fat diet

## Abstract

The signaling of membrane receptors is modified in obesity characterized by low-grade inflammation. The obesity-resistant state of organisms is poorly understood. We analyzed the generation of reactive oxygen species (ROS) initiated though membrane formyl peptide receptors (Fpr1, Fpr2) in bone-marrow granulocytes of obesity-resistant mice (ORM). A chemiluminescence assay was used to assess NADPH-oxidase-related intensity of ROS generation. ORM were chosen from animals that received high-fat diets and had metric body parameters as controls (standard diet). High spontaneous ROS production was observed in ORM cells. The EC50 for responses to bacterial or mitochondrial peptide N-formyl-MLF was higher in ORM with and without inflammation vs. the same control groups, indicating an insignificant role of high-affinity Fpr1. Increased responses to synthetic peptide WKYMVM (Fpr2 agonist) were observed in controls with acute inflammation, but they were similar in other groups. Fpr2 was possibly partially inactivated in ORM owing to the inflammatory state. Weakened Fpr1 and Fpr2 signaling via MAPKs was revealed in ORM using specific inhibitors for p38, ERK1/2, and JNK. P38 signaling via Fpr2 was lower in ORM with inflammation. Thus, a high-fat diet modified FPRs’ role and suppressed MAPK signaling in NADPH-oxidase regulation in ORM. This result can be useful to understand the immunological features of obesity resistance.

## 1. Introduction

Obesity, a precursor to type 2 diabetes mellitus, as well as cancer, arthritis, hypertension, stroke, and myocardial infarction, is characterized by low-grade chronic inflammation and modified expression and signaling of membrane receptors, which is caused by genetic and epigenetic factors [1,2,3]. First of all, the insulin receptor dysfunction leading to insulin resistance (IR) is of fundamental importance and defects in the distal insulin signaling pathway (GLUT4 including) are likely drivers of IR [4]. Among numerous obesity-related genes, mutations in genes of the leptin receptor, G protein-coupled receptor 24, and melanocortin receptors 3 and 4 are argued to be the main factors of the development of monogenic obesity in humans [5]. Genes implicated in severe and early-onset obesity-coding receptors or signaling components including LEPR (human, mice), MC4R, NTRK2, SIM1, SH2B1, MRAP2, KSR2, and ADCY3 (human) were revealed [3].

Obese inflammatory conditions in different tissues are triggered and maintained by such epigenetics factors as dietary components (fatty acids including), hypoxia, danger/pathogen-associated molecular patterns (DAMP/PAMPs–LPS, C-type lectins, N-formyl peptides, structure fragments of pathogens, and DNA), and intestinal microbiota (alterations of which can lead to metabolic endotoxemia owing to raised plasma levels of LPS) [6,7]. The inflammatory state evinces increased serum concentrations of C-reactive protein, and is accompanied by an excessive level of proinflammatory cytokines (TNF-α, IL-1β, IL-6, IL-8) and decreased production of regulators with anti-inflammatory properties (IL-10, IL-4, TGF-β, prostaglandin E2, factors controlling Toll-like receptors, and NF-κB signaling) [2,8]. Epigenetic regulation of Toll-like receptors (TLRs–TLR2 and TLR4) via changes in the gut microbiota correlated significantly with body mass index [1]. Murine high-fat diet (HFD) induced obesity led to elevated blood and intestinal levels of the microbiota-produced N-formyl peptide fMLF, a pro-inflammatory agent with multiple targets in an organism [9]. Immune cells (Th1, CD8+ TEM/TE, M1 macrophages) contribute to obesity-linked IR secreting type 1 cytokines (IFNγ, TNF-α, IL-1β, IL-6, IL-12). Additionally, neutrophils participate, releasing elastase, myeloperoxidase, and IL-1β [6].

Formylated peptides, the source of which are bacteria and mitochondria of broken cells, play a role in the inflammatory process in obesity, acting through membrane formylpeptide receptors (FPRs). The mouse FPR gene family consists of eight members (Fpr1–Fpr3 and Fpr-rs3–Fpr-rs7) that localize to chromosome 17A3.2 [10]. Fpr1 and Fpr2, receptors with high- and low-affinity to fMLF, accordingly, are predominantly expressed in mouse neutrophils. Their roles in inflammation are different: FPR1 is considered to be the primary receptor for detection of N-formyl peptides, whereas FPR2 (69% homology to FPR1) can bind to a variety of ligands (formyl peptides, annexin A1, serum amyloid A, lipoxin A4, and others) [8,9,11]. FPR2 mediates both inflammatory and anti-inflammatory functions, suggesting ligand-dependent biased signaling; it is considered as a therapeutic target in inflammation, in particular for bowel disease [8]. Both receptors have been attributed a role in obesity: the FPR1 gene was downregulated in obese humans, and CYBB gene-coding NADPH-oxidase membrane subunit gp91phox was also downregulated in obese humans [12]; genetic or pharmacological inhibition of Fpr1 increased insulin levels and improved glucose tolerance, dependent upon glucagon-like peptide 1 [9]; the lack of ALX/FPR2 led to the development of spontaneous obesity, reduced life span, amplified leukocyte dysfunction, and facilitated profound interorgan non-resolving inflammation [13]. Bone marrow transplantations between wild-type and Fpr2-/- mice and myeloid-specific Fpr2 deletion demonstrated that Fpr2-expressing myeloid cells exacerbated HFD-induced obesity, IR, glucose/lipid metabolic disturbances, and inflammation [14].

The obesity-resistant (OR) state of an organism is of great interest to medicine because the determination of the reason for an OR state will provide new possibilities to develop effective strategies against obesity-related metabolic disorders. However, the OR state is poorly understood and its causes have not been identified. Studies of the metabolic characteristics of obesity resistance are carried out using OR rodent species including knockout mice, mice of strains prone (C57BL/6j) or resistant (db/db, A/J, BALB/c) to obesity, and obesity-prone (Osborne–Mendel, Sprague–Dawley) and obesity-resistant S5B/Pl (S5B) strains [15]. HFD-fed animals are categorized as obesity-prone or OR, based on body weight versus lean controls. Some biomarkers and biomedical factors affecting obesity resistance were revealed: higher alkaline phosphatase level and α-glucosidase activity, lower expression of CD36 on the tongue, changed expression level of olfactory receptor mRNA, an increase in the enzymatic activities linked to lipid metabolism in the small intestine, greater orexin signaling, and spontaneous physical activity; selectively enhanced sympathetic activity and reduced noradrenergic activity; and behavioral factors including physical activity and sleep [15,16,17,18]. As for genes, knockout of the response gene to complement 32 (RGC32) and vimentin genes resulted in less adiposity and weight, improved IR and glucose tolerance, and ameliorated systemic inflammation in HFD-consuming mice, thus forming an OR phenotype [19,20]. In contrast, saved active TLR5 promotes OR because its gene loss alters the gut microbiota and induces low-grade inflammatory signaling, which might in turn cross-desensitize insulin receptor signaling and other aspects of metabolic syndrome (hyperlipidemia, hypertension, insulin resistance, and increased adiposity) [21]. OR animals fed HFD were believed to have a gut microbiota different from the microbiota of animals prone to obesity [22]. Ruminococcaceae_UCG-013 was identified as a novel microbiome biomarker for obesity resistance [23]. The genomic loci rendering resistance to obesity in macaques chronically consuming HFD were identified, and are as follows: three single-nucleotide polymorphisms—two in apolipoprotein B (APOB) and one in phospholipase A2 (PLA2G4A) [24].

There are a small number of articles concerning the analysis of immune reactions in OR, particularly expression and functionality of membrane receptors. It was revealed in obese-resistant BALB/c mice that restrictive HFD feeding compared with normal diet led to significantly higher body weights, visceral fat mass, and plasma interferon-γ concentrations which were associated with changes in the frequencies of granulocytes and NK cell subsets as well as in the surface expression of NK cell maturation markers and receptors [25]. It was noted that diet-induced obesity resulted in alterations in immune and metabolic profiles that are distinct from effects caused by HFD alone [26]. It was also revealed that Fpr2 deficiency reduced body weight gain in mice fed HFD and inhibited macrophage infiltration and M1 polarization in mice with diet-induced obesity. reducing tissue and systemic inflammation [14]. However, the behavior and functioning of granulocytes, the cells of myeloid origin, and one of the key players in obesity-induced inflammation have not been studied in obesity resistance.

The aim of the work was to investigate NADPH-oxidase-dependent generation of reactive oxygen species (ROS) initiated though membrane formyl peptide receptors (Fpr1, Fpr2) in bone-marrow granulocytes of obesity-resistant mice (ORM) fed high-fat diets (HFD). As long-term HDF consumption is a stressor factor leading to the development of obesity that is accompanied by low-grade chronic inflammation and modified immune profiles in obesity-prone mice, we assumed that in ORM it would also modify immune cell functions including ROS production initiated by membrane FPRs. Given the important role of FPRs in inflammation, it was of interest to find out whether further changes in the investigated granulocyte function occur during an acute inflammatory reaction that exacerbates the pathological process.

## 2. Materials and Methods

### 2.1. Animals

C57BL/6j, male mice (4 weeks, 10–15 g) were obtained from the Animals Breeding Center (the Branch of Shemyakin-Ovchinnikov Institute of Bioorganic Chemistry, Russian Academy of Sciences, Pushchino, Russia). Six-week-old mice were randomized to either high-fat or normal (standard) diets ad libitum for 16 weeks (the experimental or control groups, accordingly). A standard granular feed (complete feed for mice and rats, Mucedola Standard Diet 4RF21, Italy) was used for the control group. The energy value was 260 kcal/100 g. For the experimental group, a standard feed was used with the addition of melted pork fat (lard) with the content of the main nutrients in the diet at the rate of 45% fat, 35% carbohydrates, 20% protein, and a total calorie content of 516 kcal/100 g, as well as table salt and sodium glutamate to improve taste qualities. After 16 weeks, the HFD-feeding mice whose body lengths and weights did not differ from controls, were included in the experimental group (obesity-resistant mice, ORM) (Table 1). The mice fed HFD and having significantly higher body weight gain than controls were not taken in the experimental group. Then, four groups were formed: (I) controls; (II) controls with acute inflammation; (III) ORM; and (IV) ORM with acute inflammation. Acute inflammation was caused by intraperitoneal injection of zymosan suspension in Hanks’ solution (5 mg/mL, 150 µL) 12 h before the experiment. Controls (I) and ORM (III) were injected with 150 µL Hanks’ solution intraperitoneally.

Fasting blood glucose concentration was measured using the glucose analyzer Satellite express (ELTA Ltd., Moscow, Russia) in whole blood obtained from the tail vein after 12 h fasting. All mice were fasted overnight in individual cages before being sacrificed. Blood samples for biochemical analysis were drawn from the inferior vena cava. Biochemical analysis was conducted using a Sapphire-400 spectrophotometer (Tokyo Boeki Ltd., Tokyo, Japan).

Animal experiments were carried out according to the National Institutes of Health Guide for the Care and Use of Laboratory Animals (1978) and the Animal Care and Use Commissions at the Branch of Shemyakin-Ovchinnikov Institute of Bioorganic Chemistry of the Russian Academy of Sciences (protocol No. 871/22) and Institute of Cell Biophysics of the Russian Academy of Sciences (protocol No. 12306, 2006).

### 2.2. Organ Isolation

Mice were weighed, anesthetized, and sacrificed. The thymus, spleen, and liver of each animal were removed and weighed. The organ mass indexes were calculated as the ratio of the weight of each organ to the body weight of the animal (Table 1).

### 2.3. Cell Isolation

Granulocytes were isolated from mice bone marrow as described earlier [27,28] in Percoll gradient in PBS (55%, 62.5%, 78%, *vol/vol*). Cell viability was ~98% determined by trypan blue staining. The purity and maturity of granulocytes were performed >85% by Hoechst 33342 staining and anti-Gr-1 antibody (PE anti-mouse Ly-6G/Ly-6C (Gr-1) antibody, cat. 108,407, Biolegend), respectively, and imaging by a Leica DM2500 microscope (Leica, Wetzlar, Germany). The granulocyte suspension (107 cells/mL) was stored in Ca^2+^-free Hanks’ balanced salt solution (HBSS, pH 7.35–7.4) for 1 h at 4 °C before the experiment.

### 2.4. Reagents

Isolated granulocytes were stimulated by formyl peptide N-formyl-Met-Leu-Phe (fMLF, 0.1–50 µM) and the synthetic hexapeptide WKYMVM, Fpr2 agonist (0.1–50 µM). Inhibitors of PLC (U73122, 0.2 and 2 µM), PKC (GF109203X, 1 µM), p38MAPK (SB202190, 10 µM), ERK1/2 (FR180204, 10 µM), and JNK (SP600125, 10 µM) [28,29,30,31,32] were used to establish whether the role of indicated enzymes was modified in the signal transduction from FPRs to NADPH-oxidase in ORM. Cells were incubated with one of the inhibitors for 20 min before activation. All reagents were purchased from Sigma-Aldrich (St. Louis, MO, USA).

### 2.5. Generation of Reactive Oxygen Species

ROS generation was estimated using the luminol-dependent chemiluminescence (CL) technique (CHEMLUM-12 deviсe, ICB RAS, Pushchino, Russia) as described earlier [33]. Luminol was used to measure extra- and intracellular ROS including superoxide anion radical, hydrogen peroxide, hydroxyl radical, and hypochlorite [34,35,36]. Samples for measurements were prepared as follows: 169 μL HBSS with Ca^2+^ (1 mM), 20 μL granulocyte suspension (10^7^ cells/mL), 7 μL luminol (0.35 mM), 2 μL NaN_3_ (100 μM), and 2 μL horseradish peroxidase type VI (10^3^ U/L). NaN_3_ was used to inhibit myeloperoxidase (MPO), forming powerful oxidants (hypochlorous acid, chlorine, and chloramines) and exclude its contribution to respiratory burst [37]. CL intensity was recorded with 2.5 s intervals from 12 specimens consecutively at 37 °C. The samples were taken in triplicate for each condition. Spontaneous ROS generation (base level) was registered for 250 s, then cells were stimulated with fMLF or WKYMVM and their responses were recorded for 5–30 min depending on stimuli.

### 2.6. Data Processing

The amplitude of response (the maximal chemiluminescence intensity) and ROS production (integral of CL-curve for 50 s) were calculated using CHEMLUM-12 software (ICB RAS, Pushchino, Russia). Then data were analyzed using Matlab (MathWorks, Natick, MA, USA) software to minimize subjective bias. The effect of inhibitors was calculated as the ratio of the parameter obtained from cells treated with an inhibitor to the parameter of intact cells taken as 100%. Data are presented as mean ± SEM. ANOVA on ranks for multiple comparisons or Mann–Whitney rank sum test for 2 experimental conditions was used. The statistical significance of the difference between values was *p* < 0.05.

## 3. Results

### 3.1. Physiological and Biochemical Parameters

Body weights, thymus, and spleen mass indexes did not differ between controls and ORM (Table 1). The liver mass index was significantly lower in ORM than in controls (Table 1). Levels of liver enzymes ALT and AST were higher and LDL was lower in ORM versus controls, which indicated the presence of an inflammatory process in the liver and the destruction of liver cells (Table 1). Creatinine level was lower in ORM versus controls, indicating a decrease in the filtering ability of the kidneys (Table 1). Glucose level was higher in the blood of ORM versus controls (Table 1). 

### 3.2. Ligands of Formyl Peptide Receptors (FPRs) and ROS Generation

Granulocytes undergoing myeloid differentiation in the bone marrow enter the bloodstream as needed and, upon receiving DAMPS/PAMPS signals, migrate through a concentration gradient of attractants to the inflammatory center, where they realize their defense functions [38,39]. Operating through membrane receptors on the cell surface belonging to the FPR family, N-formyl peptides are a driver of migration and activators of different granulocyte functions among which ROS generation is one of the most important. Granulocytes regulated through various receptors are inducers of inflammation and participants of its completion or transition to the chronic stage in pathologies [8,40,41,42,43]. Based on the abovementioned, we assumed that in HFD-consuming ORM the ability of granulocytes to produce ROS in response to activation of FPRs is modified under basic conditions and with the development of acute inflammation.

The intensity of spontaneous ROS generation (before stimuli addition) was significantly higher in the cells of ORM both with and without acute inflammation compared with the corresponding control groups (Figure 1). Acute inflammation in both cases did not influence ROS base level, indicating that HDF promoted the amplification of ROS production by granulocytes even in the absence of obesity signs in mice, possibly supporting NADPH-oxidase in an active state or stimulating lipid and lipoprotein peroxidation [44,45,46].

Amplitude and ROS production in respiratory responses to fMLF or WKYMVM depended on their concentrations (Figure 2). Both parameters were significantly lower in response to 5 μM fMLF and did not differ at higher fMLF concentrations in ORM versus controls (Figure 2C,D). These results are consistent with the fact that the WKYMVM concentration dependences coincided in the control and OR groups (Figure 2E,F), indicating the suppression of the activity of high-affinity Fpr1.

In control mice, acute inflammation did not change the parameters of fMLF-induced respiratory burst and increased both parameters in response to 0.5 and 1 μM WKYMVM versus ones without inflammation (Figure 2). Significant differences at higher WKYMVM concentrations were not revealed. On the contrary, in ORM cells, acute inflammation did not change the parameters of WKYMVM-induced generation of ROS and decreased parameters of responses to 1–10 μM fMLF versus ones in the corresponding group without inflammation (Figure 2). It was also noted that responses to both agonists were significantly lower in both ORM groups versus controls with acute inflammation.

The concentration of agonist that gives a half-maximal effect (EC50), considered as a characteristic of the agonist’s efficacy [47], was calculated for fMLF and WKYMVM in all groups (Table 2). EC50 for fMLF was significantly higher in ORM without and with inflammation compared with the same control groups; the highest value of EC50 was in ORM with inflammation. These results indicated the weakened functionality of high-affinity Fpr1 in ORM and ORM with acute inflammation. EC50 for WKYMVM was similar in all groups (Table 2). Thus, fMLF (activating both Fpr1 and Fpr2), but not WKYMVM (Fpr2 agonist), makes it possible to distinguish controls from ORM, as well as ORM from ORM with acute inflammation. The WKYMVM signal allowed us to distinguish acute inflammation in controls. This situation can be due to features of binding of peptides with FPRs, receptor expression, state (e.g., desensitization in ORM), or signaling [8,12,48,49,50,51]. Next, we analyzed FPR signaling to NADPH-oxidase with the participation of PLC, and PKC, as the main signaling components of these receptors, as well as MAPKs, which fine-tune the activity of the oxidase and participate in pathological processes.

### 3.3. PLC and PKC Signaling in FPR-Mediated Respiratory Burst

Incubation of the cells with 0.2 µM U73122, PLC inhibitor led to a slight inhibition of responses to 1 µM fLMF and 1 µM WKYMVM in all groups; the effects were similar in controls and ORM with and without inflammation (Figure 3). Higher U73122 concentration (2 µM) strongly inhibited responses to FPR ligands; its effects were stronger in ORM compared to controls (Figure 3). Acute inflammation did not change the effects of 2 µM U73122 on fMLF-induced ROS generation in all groups and enhanced suppression of the response to WKYMVM only in controls (Figure 3). These data indicated that HFD and acute inflammation led to a gain of the positive regulation of NADPH-oxidase activity by PLC.

The PKC inhibitor GF109203X (1 µM) significantly suppressed Fpr1- and Fpr2-mediated responses in all groups (Figure 4). Its effects on fMLF-initiated respiratory burst were weakened in ORM versus controls and unchanged in other groups (Figure 4). The effects of GF109203X on WKYMVM-induced respiratory burst were attenuated in controls with inflammation versus the same group without inflammation (Figure 4). The results demonstrated the decreased role of PKC in the regulation of NADPH-oxidase activity initiated via FPRs in ORM granulocytes, and also in the cells of controls with inflammation activated via Fpr2.

### 3.4. Involvement of MAP Kinases in FPR-Mediated Respiratory Burst

Inhibitory analysis showed that SB202190 and SP600125, inhibitors of p38MAPK and JNK, accordingly, significantly suppressed respiratory burst induced by 1 µM fMLF in granulocytes of all mice groups (Figure 5A,B). However, their inhibition was significantly lower in ORM than in control mice (Figure 5A,B). Acute inflammation did not change the effects of SB202190 and SP600125 in control mice but it weakened the effect of SB202190 in ORM versus the corresponding group without inflammation (Figure 5A,B). FR180204, an inhibitor of ERK1/2, slightly suppressed the fMLF-induced respiratory response in control mice with and without inflammation and did not change it in ORM in both mice with and without (except ROS production) acute inflammation (Figure 5A,B).

SB202190 suppressed respiratory burst initiated by 1 µM WKYMVM in all groups of mice (Figure 5C,D). Acute inflammation did not change SB202190′s effect in controls, but in ORM with inflammation, this inhibition was stronger versus the corresponding mice without inflammation (Figure 5C,D). SP600125 and FR180204 decreased parameters of responses to WKYMVM in granulocytes of control mice and did not alter it in ORM (Figure 5C,D). Acute inflammation did not change their effects in controls and strengthened them in ORM versus the corresponding group without inflammation (Figure 5C,D). The results indicated a significant modification (weakening) of Fpr1-initiated p38MAPK and JNK signaling to NADPH-oxidase in ORM and stronger weakening p38MAPK and ERK signaling in ORM with acute inflammation.

## 4. Discussion

We investigated the physiological and biochemical characteristics of obesity-resistant mice (ORM) and functional parameters of their bone marrow granulocytes, namely NADPH-oxidase-mediated ROS generation in resting cells and cells stimulated by agonists of membrane formyl peptide receptors (Fpr1, Fpr2). It was revealed that (1) liver mass index levels, LDL, were lower while ALT and AST levels were higher in ORM versus controls; (2) spontaneous ROS production was higher in ORM cells with and without inflammation; and (3) kinetic parameters of fMLF-induced ROS generation were lower and EC50 for these responses was higher in ORM with and without inflammation versus the corresponding control groups. The responses to WKYMVM (Fpr2 agonist) were higher in controls with inflammation, but they were similar in other groups; (4) PLC and PKC inhibition suppressed Fpr1 and Fpr2 signal transduction to NADPH-oxidase in controls and ORM cells. However, Fpr1 and Fpr2 signaling via PLC was enhanced, while Fpr1 signaling via PKC was attenuated in ORM granulocytes. Acute inflammation enhanced the effect of the PLC inhibitor and weakened the effect of the PKC inhibitor on Fpr2-mediated ROS generation in controls’ cells and did not change them in ORM cells; and (5) p38 and JNK signaling triggered by Fpr1 was weakened in ORM granulocytes with and without inflammation, whereas Fpr2-mediated p38 signaling was enhanced only in ORM with inflammation.

Nutrition disorders (obesity and malnutrition) and also stress, aging, and infectious processes act on the thymus and spleen, which are hematopoietic organs with actively proliferating tissue with high sensitivity and the ability to respond quickly to environmental factors [52,53]. The liver plays an important role in the regulation of fat and carbohydrate metabolism, so long-term HFD consumption may change its function [54]. In this study, we estimated the mass indexes of these organs and did not reveal differences of thymus and spleen mass indexes between controls and ORM (Table 1). However, we showed a decreased liver mass index in ORM that, when combined with increased levels of ALT and AST as well as a decreased LDL level, indicates the presence of an inflammatory process in the liver and its impairment in ORM. These data are partially consistent with the results of other authors who also demonstrated higher ALT and AST levels in HFD-fed ORM versus controls, although their liver masses did not differ [55]. It was also shown that high levels of fatty acids, particularly saturated fatty acids such as palmitate, stimulated hepatoxicity, metabolic dysfunction, and subsequent hepatic inflammation [56].

The spontaneous level of ROS generation was higher in granulocytes of ORM without and with acute inflammation compared with the same controls, indicating amplification of ROS production by unstimulated granulocytes of HFD-fed ORM (Figure 1). The increased intensity of spontaneous ROS generation may be caused by hyperglycemia because higher glucose levels were observed in the blood of ORM versus controls (Table 1), which can affect the phagocyte state [57,58,59]. Our results are consistent with the data of other authors who also revealed higher glucose levels in the blood of ORM versus controls, but the level was significantly lower in obesity-prone mice [55]. The authors claimed that according to biochemical parameters and lipidomic profiling of liver tissue, mice in the OR group fed HFD were closer to those in the group with a normal diet compared with those in the obesity-prone group [55].

Then, we assessed the activation of membrane NADPH-oxidase-dependent ROS generation via Fpr1 and Fpr2 on bone marrow granulocytes in ORM (Figure 2). Parameters of respiratory burst were lower in ORM to 5 µM fMLF and in ORM with inflammation to 1 µM fMLF; the EC50 for these responses was higher in ORM without and with inflammation versus the same controls. Responses to high concentrations of fMLF (activating both Fpr1 and Fpr2) and to Fpr2 agonist WKYMVM did not differ between groups, indicating an insignificant role of Fpr1 in this concentration range of fMLF. Acute inflammation increased Fpr2-mediated ROS production in controls (Figure 2), indicating the involvement of low-affinity FPRs in the inflammation process [8,51]. Decreased responses to 1–10 μM fMLF and the absence of differences in WKYMVM responses in ORM with inflammation versus those without inflammation may be explained by the changed state of Fpr2 (including their quantity on the cell membrane and signaling) in ORM owing to their inflammatory state. This could be one of the reasons for obesity resistance in response to HFD. It was shown that Fpr2 on the immune cells involved in diet-induced obesity, metabolic disturbances, and inflammation [14]. The authors revealed that Fpr2 deficiency reduced body weight gain in HFD-fed mice through enhancing energy expenditure, especially in skeletal muscle, and also reduced tissue and systemic inflammation in mice with diabetic-induced obesity by inhibiting macrophage infiltration and M1 polarization [14]. The authors also found that myeloid Fpr2 played an important role in diabetic-induced obesity, IR, and glucose/lipid dysmetabolism [14].

Agonist binding by FPRs causes the dissociation of heterotrimeric Gαi-βγ protein complex to Gα- and Gβγ-subunits and further divergence of the signal transduction that creates a complex network, regulating the phosphorylation of NADPH-oxidase subunits [60]. One of these signaling pathways is the activation of phospholipase C (PLCβ) that hydrolyzes phosphatidylinositol bisphosphate (PIP2) into inositol trisphosphate (IP3) and diacylglycerol (DAG). Then, IP3 promotes a transient Ca^2+^ release from intracellular stores. Ca^2+^ and DAG activate protein kinase C (PKC) that phosphorylates NADPH-oxidase subunits. The phosphorylation of subunits is essential for their translocation to the membrane and the assembly of the NADPH-oxidase complex to produce superoxide [60]. Earlier, PLC/PKC-dependent signaling from Fpr1 and Fpr2 to NADPH-oxidase was shown in the bone marrow granulocytes of BALB/c mice, although the Fpr2-mediated response was less sensitive to PLC and PKC inhibition compared with Fpr1-induced respiratory burst [28]. Our results verified the involvement of PLC and PKC in FPR-mediated ROS production in the bone marrow granulocytes of C57BL/6j mice. In ORM granulocytes, the inhibition of PLC led to greater suppression of the responses compared with controls, indicating stronger positive regulation of FPR-mediated respiratory burst via PLC in ORM (Figure 3). PKC inhibition weakened Fpr1- and Fpr2-mediated responses in all groups, but in ORM the effect of the PKC inhibitor was weaker than in controls and significantly differed in the case of the response to fMLF mediated mainly by Fpr1 (Figure 4A,B). This result was unexpected since the increase in PLC activity leads to the increase in PKC activity in neutrophils [61]. The inhibition of both enzymes suppressed the responses to both agonists, but the change of the effects of PLC and PKC inhibitors were opposite in ORM versus their effects in controls (Figure 2 and Figure 4). One of the possible explanations may be modified Ca^2^ release from internal sources in response to IP3 and/or the involvement of Ca^2+^-independent isoforms of PKC in FPR-mediated NADPH-oxidase activation in ORM, which requires further investigations [62,63,64].

HFD increases the levels of saturated fatty acid that induce stress-sensitive MAPK pathways. MAPKs cause phosphorylation of nuclear and cytoplasmic substrates, leading to adaptation to the new diet by regulating insulin signaling, blood glucose concentration, and obesity [65]. Obesity and the associated chronic inflammation result in the release of pro-inflammatory cytokines activating the stress-responsive MAPKs such as p38, JNK, and ERK1/2 [65,66]. MAPKs were shown to be critical effectors in physiological and pathophysiological hepatic inflammation [66]. In our study, the involvement of MAPKs was estimated in the regulation of membrane NADPH-oxidase via FPRs on bone marrow granulocytes in ORM using specific MAPK inhibitors of p38 (SB202190), ERK1/2 (FR180204), and JNK (SP600125). It should be noted that the specificity of some inhibitors including the specificity of SP600125 to JNK is discussed in the literature. Bennett and colleagues demonstrated in cell-based assays that SP600125 blocks JNK and no other inflammatory-signaling cascades, and seems to have a selectivity of at least 10-fold for the JNK pathway [67]. It was also noted that some ATP-competitive inhibitors of JNK, such as SP600125 and AS601245, are widely used in vitro; however, this type of inhibitor lacks specificity as it indiscriminately inhibits the phosphorylation of all JNK substrates [6]. The latter conclusion does not contradict the first. Nevertheless, these inhibitors are widely used to assess the involvement of the corresponding kinases in various processes, including ROS generation by neutrophils [28,29,30,31]. In the study, we revealed weakened Fpr1 signaling predominantly via p38 and JNK in ORM granulocytes and enhanced Fpr2 signaling via p38 in ORM with inflammation versus ORM. ERK1/2 was involved only slightly in FPR-mediated NADPH-oxidase regulation and this did not differ between controls and ORM (Figure 5). Thus, p38 and JNK were less involved in FPR-mediated respiratory burst by granulocyte NADPH-oxidase in ORM, possibly because of decreased activity or partial inactivation of these MAPKs that can protect from excessive ROS production observed in obesity-prone and diabetic mice [68]. Acute inflammation led to stronger inhibition of p38 MAPK via Fpr2 in ORM, indicating a significant role of p38 in inflammatory processes in ORM. Our data can be explained by the fact that p38 MAPK and JNK are considered as pro-inflammatory factors, the activation of which promotes inflammation and the development of IR that was confirmed in other cells and tissues [65,69,70,71]. In particular, the participation of p38α and p38δ was revealed in IR, oxidative stress-induced β-cell failure and hepatic gluconeogenesis [69,70,72]. In murine models of obesity, hepatic p38 MAPK was activated and when p38 MAPK was overexpressed in the liver, it led to the impairment of insulin signaling [73,74]. P38δ was found to be elevated in the liver of obese patients, while mice lacking p38γ/δ in myeloid cells were resistant to diet-induced fatty liver, hepatic steatosis, and glucose intolerance [75]. The authors explained this protective effect by defective migration and hepatic infiltration of p38γ/δ-deficient neutrophils [75]. Other authors revealed the resistance of macrophages lacking p38α in the development of steatohepatitis in response to high-fat/high-cholesterol diets [71]. Derived from macrophage p38α-deficient mice, the primary hepatocytes showed decreased steatosis and inflammatory damage through reduced secretion of pro-inflammatory cytokines (TNF-α, CXCL10, and IL-6), which regulate M1 macrophage polarization [66,71]. The JNK pathway is also required for HFD-induced obesity and IR [76]. JNK activity was revealed to be elevated in liver, muscle, and adipose tissues in obesity. The absence of JNK1 led to decreased adiposity, significantly improved insulin sensitivity, and enhanced insulin receptor signaling capacity in dietary and genetic (ob/ob) models of mouse obesity [76]. JNK can promote inflammation by polarizing macrophages to the M1 inflammatory phenotype, as is required for M1, but not M2, macrophage development [65,77]. The loss of JNK function in murine models was observed to protect mice against the development of IR when fed HFD [78]. Thus, MAPKs can be potential targets for the treatment of metabolic syndrome and prediabetes due to their key role in the support of energy homeostasis and normal glycemia [78].

## 5. Conclusions

Obtained data confirm our hypothesis that despite the similarity of many physiological and biochemical parameters between mice on a standard diet and obesity-resistant mice fed high-fat diets [26,55], a high-fat diet still changes the functioning of some organs and systems, including the liver and immune system, and leads to the development of chronic inflammatory processes and subsequent aggravated disturbances of their functions. The regulatory function of FPRs in inflammation and obesity, and also their functional duality depending on the nature of the ligand were defined. In this study, for the first time, we demonstrated modified FPR-mediated reactions of innate immune cells such as granulocytes and that altered signaling from membrane receptors to NADPH-oxidase initiated two FPR agonists in obesity resistance. It remains unclear whether the observed abnormalities are due to a congenital feature of an organism or the action of specific nutrition components and the consequence of a biochemical shift. In addition, it is extremely important to figure out the basis of obesity resistance for the human “overeating” community. Nevertheless, our data can be useful to understand immunological features of obesity resistance and open the possibility of using of FPRs as potential therapeutic targets to inhibit obesity-related inflammation and to develop strategies against obesity-related metabolic disorders.

## Figures and Tables

**Figure 1 membranes-13-00306-f001:**
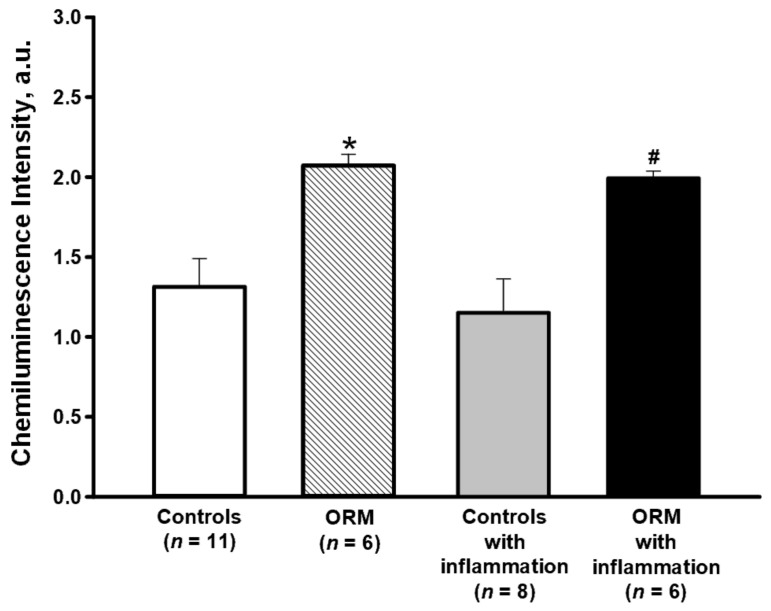
Spontaneous generation of reactive oxygen species in bone marrow granulocytes of controls and obesity-resistant mice (ORM) with or without acute inflammation. Acute inflammation was initiated by intraperitoneal injection of 150 µL zymosan suspension (5 mg/mL). Mice without inflammation were injected with 150 µL Hanks’ solution intraperitoneally. Data are presented as mean ± SEM. Significant differences between groups by ANOVA on ranks: * *p* < 0.05 compared to controls; # *p* < 0.05 compared to the control group with inflammation.

**Figure 2 membranes-13-00306-f002:**
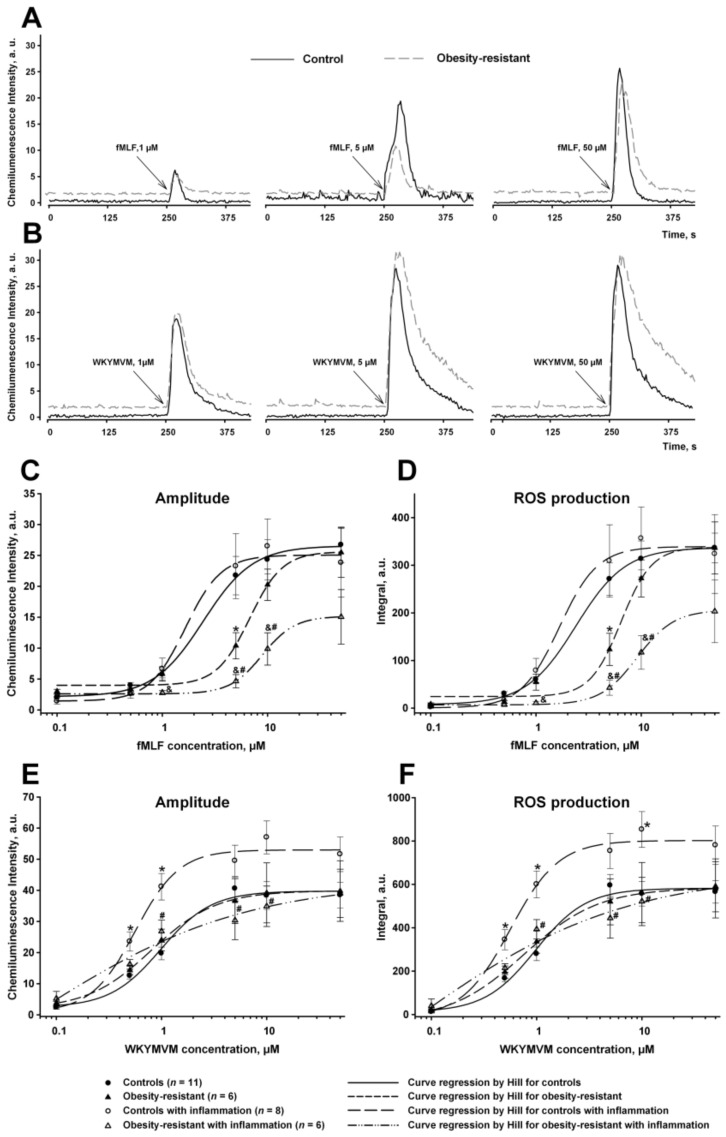
Dependences of the amplitude of response and the production of reactive oxygen species (ROS) on the concentrations of FPR agonists. (**A**,**B**) Experimental records of cell responses to fMLF and WKYMVM (1, 5, 50 µM), respectively. (**C**,**D**) Kinetic parameters of fMLF-induced respiratory burst. (**E**,**F**) Kinetic parameters of WKYMVM-induced respiratory burst. Data are presented as mean ± SEM. Significant differences of parameters by ANOVA on ranks: * *p* < 0.05 compared to controls; # *p* < 0.05 compared to controls with inflammation; and & *p* < 0.05 between obesity-resistant mice with and without inflammation.

**Figure 3 membranes-13-00306-f003:**
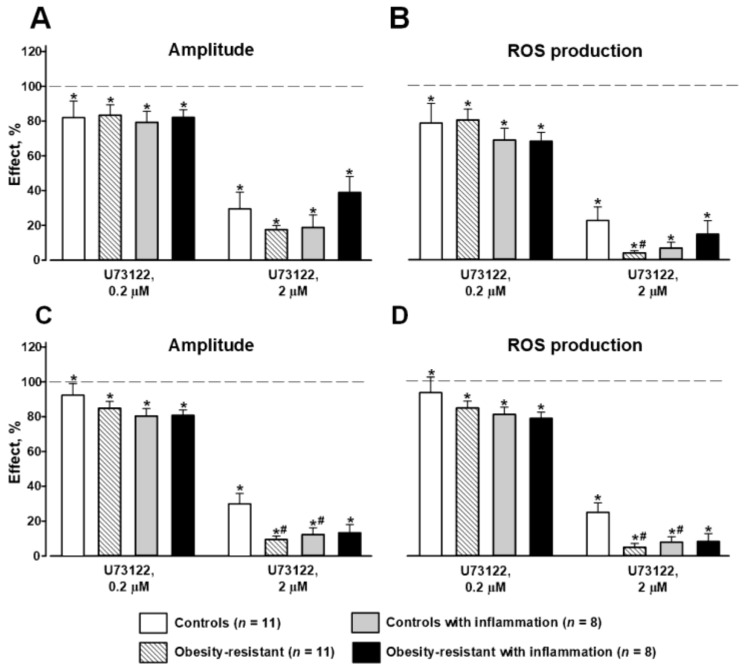
Effects of U73122, PLC inhibitor, on FPR-mediated respiratory response in murine bone marrow granulocytes. Amplitude (**A**,**C**) and reactive oxygen species (ROS) production (**B**,**D**) in responses to 1 µM fMLF (**A**,**B**) and 1 µM WKYMVM (**C**,**D**). Data are presented as mean ± SEM. Significant differences are indicated as follows: * *p* < 0.05 between parameters of inhibitor-treated and intact cells within the same group by Mann–Whitney rank sum test; and # *p* < 0.05 compared to controls by ANOVA on ranks.

**Figure 4 membranes-13-00306-f004:**
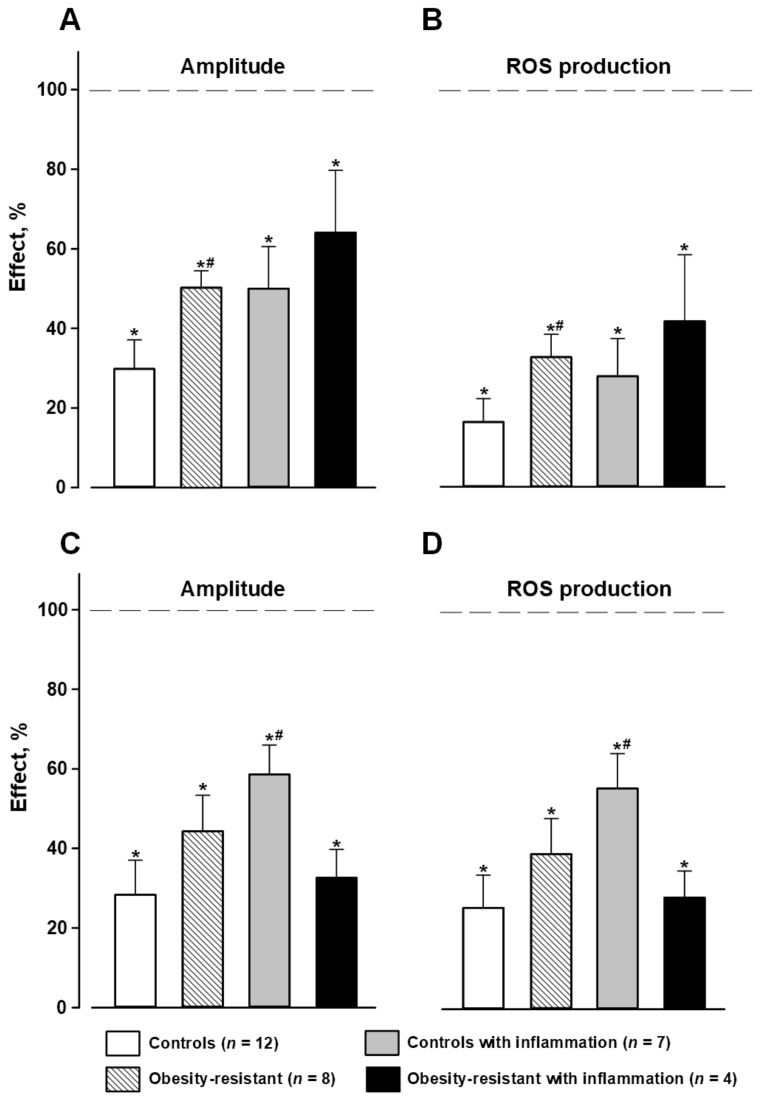
The effects of the inhibitor of PKC on the FPR-mediated respiratory response in murine bone marrow granulocytes. Amplitude and production of reactive oxygen species occurred (ROS) in response to 1 µM fMLF (**A**,**B**) and 1 µM WKYMVM (**C**,**D**). GF109203X (1 µM) was used to inhibit PKC. Data are presented as mean ± SEM. Significant differences of parameters are indicated as follows: * *p* < 0.05 between inhibitor-treated and intact cells within the same group by Mann–Whitney rank sum test; and # *p* < 0.05 compared to controls by ANOVA on ranks.

**Figure 5 membranes-13-00306-f005:**
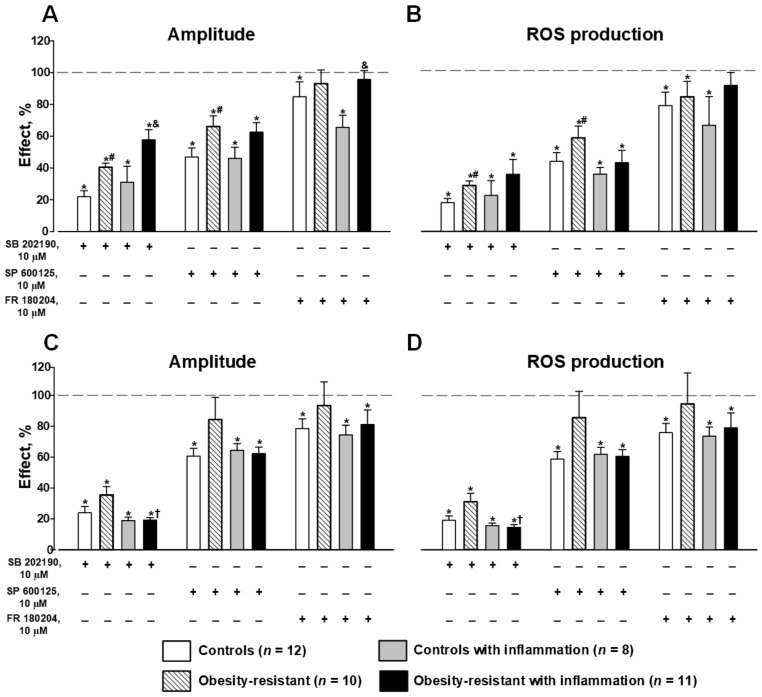
The effects of inhibitors of MAP kinases on FPR-mediated respiratory response in murine bone marrow granulocytes. Amplitude and production of reactive oxygen species (ROS) in responses to 1 µM fMLF (**A**,**B**) and 1 µM WKYMVM (**C**,**D**). Data are presented as mean ± SEM. Significant differences of parameters by Mann–Whitney rank sum test are indicated as follows: * *p* < 0.05 between inhibitor-treated and intact cells within the same group; by ANOVA on ranks: # *p* < 0.05 compared to controls; and & *p* < 0.05 compared to controls with inflammation; † *p* < 0.05 compared to obesity-resistant mice.

**Table 1 membranes-13-00306-t001:** Baseline characteristics of mice in the control and obesity-resistant groups.

	Controls, *n* = 9	Obesity-Resistant, *n* = 12
Physiological characteristics		
Body weight, g	27.7 ± 0.7	29.1 ± 0.6
Thymus mass index, %	0.16 ± 0.03	0.14 ± 0.01
Spleen mass index, %	0.30 ± 0.03	0.37 ± 0.03
Liver mass index, %	6.62 ± 0.35	3.63 ± 0.22 *
Biochemical blood analysis		
Glucose, mmol/L	7.43 ± 0.52	10.42 ± 0.54 *
Total cholesterol, mmol/L	1.80 ± 0.01	2.17 ± 0.12
HDL, mmol/L	0.95 ± 0.05	1.08 ± 0.07
LDL, mmol/L	0.56 ± 0.05	0.13 ± 0.01 *
Triglycerides, mmol/L	0.41 ± 0.06	0.54 ± 0.03
ALT, U/L	21.7 ± 3.4	89.2 ± 30.8 *
AST, U/L	41.7 ± 8.6	141.5 ± 0.2 *
Total bilirubin, µmol/L	3.2 ± 1.3	1.70 ± 0.30
Creatinine, µmol/L	54.0 ± 2.6	32.4 ± 4.0 *
Urea, mmol/L	11.5 ± 1.0	8.4 ± 1.1

Notes. ALT, alanine aminotransferase; AST, aspartate aminotransferase. * indicates significant differences between groups according to the Mann–Whitney rank sum test, *p* < 0.05.

**Table 2 membranes-13-00306-t002:** EC50 values for FPR ligands initiated respiratory burst in murine bone marrow granulocytes.

	EC50, µM fMLF-Induced Response		EC50, µM WKYMVM-Induced Response	
	Amplitude	ROS Production	Amplitude	ROS Production
Controls	2.44 ± 0.25	2.39 ± 0.13	0.99 ± 0.19	0.98 ± 0.18
ORM	6.76 ± 0.78 *	6.44 ± 0.82 *	0.81 ± 0.04	0.76 ± 0.05
Controls with inflammation	1.65 ± 0.38	1.67 ±0.34	0.56 ± 0.10	0.56 ± 0.09
ORM with inflammation	8.94 ± 0.23 #	9.19 ±0.28 #	0.07 ± 0.69	0.06 ± 0.73

Notes. ORM, obesity-resistant mice. Significant differences by ANOVA on ranks: * *p* < 0.05 compared to controls; # *p* < 0.05 compared to ORM.

## Data Availability

The original data are available from the corresponding author on reasonable request.

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
