# Peer review of "Modified Signaling of Membrane Formyl Peptide Receptors in NADPH-Oxidase Regulation in Obesity-Resistant Mice"

_membranes, 2023, doi:10.3390/membranes13030306_

Round 1

Reviewer 1 Report

In this manuscript, Tikhonova and colleagues showed in a model of obesity-resistance HFD-fed
mice with acute inflammation that FPR mediated signaling through NAPDH oxidase activation
in granulocytes exacerbated metabolic disturbances, and inflammation. This manuscript clearly
states the author's aim and provides a well summarized Introduction, the Results are reliable.
The abstract is concise and provides sufficient information on the study. The m
anuscript is very
well 
written. However, please comment on some minor issues.

- In your hypothesis is not clear that granulocytes activity were evaluated in control animals
and with acute inflammation.

-Please, describe in Methods specifically what kind of reactive specie was measured, not
only in general terms, as many different reactive species may be involved in several molecular
mechanisms.

-In Results you report mean ± SEM. Why not SD? Maybe refer to: https://doi.org/10.1124/mol.119.118927 for guidance.

-Considering that authors highlighted the role of NADPH-oxidase subunit NOX2, I would like
to see granulocytes derived RS-generation in the presence of a selective NOX2 inhibitor.

-Please consider include a graphic abstract.

Author Response

Point 1: In your hypothesis is not clear that granulocytes activity was evaluated in control animals and with acute inflammation.

Response 1: We expanded our hypothesis and explained the necessity of using the acute inflammation in the following way: "As long-term HDF consuming is a stressor factor leading to the development of obesity that is accompanied by low grade chronic inflammation and modified immune profile in obesity-prone mice, we assumed that in obesity-resistant mice it would also modify immune cell functions including RS production initiated by membrane FPRs. Given the important role of FPRs in inflammation, it was of interest to find out whether further changes in the investigated granulocyte function occur during an acute inflammatory reaction that exacerbates the pathological process."

Point 2: Please, describe in Methods specifically what kind of reactive specie was measured, not only in general terms, as many different reactive species may be involved in several molecular mechanisms.

Response 2: Generation of reactive species was estimated using luminol-dependent chemiluminescence (CL) technique. Luminol is known to detect extra- and intracellular reactive species including superoxide anion radical, hydrogen peroxide, hydroxyl radical, and hypochlorite [Dahlgren et al., 2008, doi:10.1007/978-1-59745-467-4_23; Vladimirov and Proskurnina, 2009; Monteseirín and Vega, 2011, doi:10.1016/j.clim.2010.12.003]. In our study we used NaN3 to inhibit myeloperoxidase (MPO) forming hypochlorous acid with the subsequent formation of chlorine and chloramines, the powerful oxidants, and exclude its contribution to respiratory burst. We added this information to the “Methods” section.

Point 3: In Results you report mean ± SEM. Why not SD? Maybe refer to: https://doi.org/10.1124/mol.119.118927 for guidance.

Response 3: We are grateful to the reviewer for the helpful reference for recommendatory settings to present experimental results “aimed at investigators in experimental pharmacology but are applicable to most fields of experimental biology”. In relation to our work, we think that when the sample size is known, it makes no crucial difference whether to use SEM or SD. However, SEM takes into account both the sample size and the variation in values relative to the mean. It also shows the probability of deviation of the mean from the population, which is important for assessing the adequacy of the sample size. SD simply shows the accuracy of the data. SD can always be obtained by multiplying SEM by the square root of the sample size. We followed this logic, as we did not find any special instructions or rules on how to present the results in the guide for the authors in the Membranes journal.

Point 4: Considering that authors highlighted the role of NADPH-oxidase subunit NOX2, I would like to see granulocytes derived RS-generation in the presence of a selective NOX2 inhibitor.

Response 4: Indeed, NOX2, is a constitutive component of phagocyte NADPH oxidase which we mention throughout the text as the enzyme that catalyzes superoxide and further other RS formation in granulocytes. However, we did not study the role of NOX2 separately in this work. NOX2 (gp91phox coding by NOX2 gene), the catalytic subunit, and 22phox, the stabilizing subunit, represent the membrane complex in resting cells and are part of the active enzyme in phagocytes. Respiratory burst initiated by FPRs develops when enzyme assembly completes from the membrane (gp91phox, p22phox) and cytoplasmic subunits (p47phox, p67phox, p40phox) as well as the small GTPase Rac1/2. In previously published work, we used apocynin, which is considered a non-selective inhibitor of NOX2 and blocks NADPH oxidase assembly (translocation of subunits to the membrane), and we observed partial or complete suppression of ROS generation depending on its concentration [Filina et al., 2022, doi:10.1016/j.bbamcr.2022.119356]. We did not analyze the role of individual subunits, including NOX2, in our submitted work. Additional methods are demanded for this purpose. We plan to use them in future work. Note that the use of selective NOX2 inhibitors is subject to certain limitations [Altenhoferet al., 2015, doi: 10.1089/ars.2013.5814].

Point 5: Please consider include a graphic abstract.

Response 5: We did a graphical abstract and added it at the end of the manuscript.

Reviewer 2 Report

The manuscript entitled “Modified signaling of membrane formyl peptide receptors in 2 NADPH-oxidase regulation in obesity-resistant mice” deals with the role of formyl peptide receptors (FPRs) in a model of obesity-resistant mice (ORM). The aim of the work is well described and also the results. For this reason, the reviewer thinks that the manuscript is suitable for publication in Membranes after minor revisions. Below some suggestions/corrections to do before publication:

-          In the introduction, please explain better the concept of biased agonism and if it’s important in this case.

-          Line 160-161, pag. 4: in paragraph 2.3, there are two references to be corrected (Boxio et al., 2004; Filina et al., 2022). Please insert the corresponding reference’s number. Filina et al., 2022 is the references [53], while Boxio et al., 2004 is not present in the references. Please insert the new bibliography in the reference section.

-          Line 181-182, pag.4 : please change “NaN3” with “NaN3

-          In the text, please change “Ca2+” with “Ca2+

-          Line 264, pag.7: PLC is repeated twice, please delete one

-          Line 388, pag.11: please change “polarization. [14].” with “polarization [14].”

-   Line 389-390, pag.11: please change “dysmetabolism, and [14].” with dysmetabolism [14].”          

Author Response

Response to Reviewer 2 Comments

Point 1: In the introduction, please explain better the concept of biased agonism and if it’s important in this case.

Response 1: The concept of biased agonism as the paradigm of ligand-based selectivity in receptor signaling is the most elaborated for G-protein coupling receptors, to which FPRs are related. Shortly, there is the ligand-dependent selectivity and effectivity for certain signal transduction pathways relative to a reference ligand at the same receptor [Smith et al., 2018, doi:10.1038/nrd.2017.229; Wisler et al., 2014, doi: 10.1016/j.ceb.2013.10.008; Hodavance et al., 2016, doi: 10.1097/FJC.0000000000000356]. The FPR family of GPCRs is particularly amenable to biased agonism [Qin et al., 2022, doi: 10.1111/bph.15919]. The authors of the cited review [Yang et al., 2023, doi: 10.1038/s41401-022-00944-0] consider this phenomenon related to FPR2 which can bind different ligands and realize biased signaling to pro-inflammatory or pro-resolving modes depending on stimuli (e.g. SAA or LXA4, accordingly). The concept was applied to human FPR1 for characteristics of RE-04-001 as a novel small molecule agonist specific for FPR1, which displays a biased signaling profile and a functional selectivity [Lind et al., 2020, doi: 10.1002/JLB.2HI0520-317R]. In accordance with the structural models, after a specific ligand binding both FPR1 and FPR2 can be stabilized in conformations that favor for initiation of any signaling, but they occur to be ineffective for other signaling pathways [Dahlgren et al., 2016, doi:10.1016/j.bcp.2016.04.014; Lind et al., 2020, doi: 10.1002/JLB.2HI0520-317R; Zhuang et al., 2022, doi: 10.1038/s41467-022-28586-0]. However, there are data conflicting with the statements: using a transgenic HeLa-FPR1 cell line and based on the bias calculations Gröper J. and colleagues did not obtain a selective activation of entirely different signaling pathways, as typically associated with biased agonists of FPR1 [Gröper et al., 2020, doi:10.3390/cells9041054]. Despite the burning question for therapy about biased agonism for FPRs, we conclude that this aspect of the functionality of FPR1 and FPR2 is not directly investigated in our work, because we do not consider the action of different ligands of the separate receptor. Therefore, we did not pay attention to it in the “Introduction” section.

Point 2: Line 160-161, pag. 4: in paragraph 2.3, there are two references to be corrected (Boxio et al., 2004; Filina et al., 2022). Please insert the corresponding reference’s number. Filina et al., 2022 is the references [53], while Boxio et al., 2004 is not present in the references. Please insert the new bibliography in the reference section.

Response 2: We are very sorry for such a mistake. We checked and corrected all reference numbers in the manuscript, added a missing reference, and inserted the new bibliography in the reference section.

Point 3: Line 181-182, pag.4: please change “NaN3” with “NaN3”.

Response 3: We corrected it.

Point 4: In the text, please change “Ca2+” with “Ca2+”.

Response 4: We corrected it.

Point 5: Line 264, pag.7: PLC is repeated twice, please delete one.

Response 5: We did it.

Point 6: Line 388, pag.11: please change “polarization. [14].” with “polarization [14].”

Response 6: We corrected it.

Point 7: Line 389-390, pag.11: please change “dysmetabolism, and [14].” with “dysmetabolism [14].

Response 7: We corrected it.

Reviewer 3 Report

The paper is interesting, but overloaded with experimental data, for correct interpretation of which additional experiments and literature references should be provided. I recommend authors to split the paper into two publications. One paper could be devoted to ROS production in ORM mice without inhibitor treatment and another one about inhibitor effects on the ROS production. Indeed, there are too many questions to this complicated study and their clarification is very difficult in a frame of one publication.

In particularly, for each used inhibitor should be provided a literature references about effect on FPR-induced signals in neutrophils. 

For example, authors concluded that Fpr1 initiate JNK signaling to NADPH oxidase (see Lines 325-326, 344) and didn’t provided no one reference about (1) activation of JNK by Fpr1 in neutrophils and  (2) role of JNK in activation of NADPH oxidase in neutrophils. As well, authors did not provide any references about using SP600125 in neutrophils in ROS assay. If it’s first time using SP600125 in luminescence-based ROS assay in neutrophils, additional experiments should be provided, including:

11)    Direct antiradical effect of SP600125 in cell-free system (enzymatic and non-enzymatic system of ROS generation).

22)    Effect of SP600125 on ROS production in neutrophils, activated via different pathways (receptors and non-receptor).

33)    Test other JNK inhibitors with different scaffold on Fpr1 in neutrophils.

44)    Test the inhibitor in cell-based non-luminescent assay.

Same experiments should be provided for other used inhibitors.

Indeed, a conclusion, based on “specificity” of a particular inhibitor could be overestimated. For example, SP600125 could inhibit other kinases.

Authors discuss (see Lines 363-366) that increased ROS generation could be caused by hyperglycemia. However, it is not at all clear how high blood glucose can affect ROS production in isolated neutrophils.

Authors discussed about different number of Fpr2 on cell membrane, but didn’t test this. Indeed, a direct determination of number of Fpr1/Fpr2 receptors on neutrophil surface in ORM in comparison with control mice should be first experiment for this study.

Minor comments:

- Please, use abbreviation ROS instead RS.

- Typo: “PLC, PLC…”  (Line 264).

Author Response

Point 1: The paper is interesting, but overloaded with experimental data, for correct interpretation of which additional experiments and literature references should be provided. I recommend authors to split the paper into two publications. One paper could be devoted to ROS production in ORM mice without inhibitor treatment and another one about inhibitor effects on the ROS production. Indeed, there are too many questions to this complicated study and their clarification is very difficult in a frame of one publication.

Response 1: We are grateful to Reviewer for interest and detailed analysis of our manuscript and suggestions. Recommendation to split the paper into two publications supplementing them with new results looks attractive and coincides with our desire to study functions of granulocytes in obesity-resistant state deeply. This work was carried out as a pilot project to study whether the production of ROS and FPR signaling to NADPH oxidase changed in obesity resistance. Really, we revealed modification of these processes. Next, we can find out the causes of such changes: expression of receptors or/and oxidase subunits, activity of signaling pathway components, etc. However, there are some limitations for experimentation with ORM now. The fact is that ORM were culled over time from an experimental group of obesity prone C57BL/ mice fed HFD for a model of metabolic syndrome and type 2 diabetes mellitus, the proportion of ORM in the whole group was nearly 20%. We cannot lay down a group of ORM eating HFD initially and get only ORM, they are a “by-product”. Moreover, ORM group with acute inflammation was included in our work for which ORM were needed that also restrict our ability for experimentation. Given the limited number of cells from an individual animal, a large reserve of animals will be needed to get representative results. Some authors, for comparison with obesity-prone C57BL/6 mice, used BALB/c mice, which are considered not inclined to obesity, but it is incorrect due to genetics differences. Other way could be a treatment of animals with any agents preventing obesity or manipulations with receptors, e.g. nicotine acetylcholine receptors [Marrero et al., 2010, doi: 10.1124/jpet.109.154633; Husson et al., 2020, doi: 10.1523/JNEUROSCI.0356-19.2020], but it would not be spontaneous obesity resistance which we studied. So it will take a long time and large animal resource to complete the work in two directions.

Point 2: In particularly, for each used inhibitor should be provided a literature references about effect on FPR-induced signals in neutrophils.

Response 2: We included the literature references about participation of the enzymes in FPR signaling which we inhibited, regulation activity of NADPH oxidase in neutrophils and usage of inhibitors in chemiluminescence-based ROS assay in neutrophils.

Point 3: For example, authors concluded that Fpr1 initiate JNK signaling to NADPH oxidase (see Lines 325-326, 344) and didn’t provided no one reference about (1) activation of JNK by Fpr1 in neutrophils and (2) role of JNK in activation of NADPH oxidase in neutrophils. As well, authors did not provide any references about using SP600125 in neutrophils in ROS assay.

Response 3: We analyzed the activation of JNK (also p38 and ERK) by Fpr1 in neutrophils from mouse bone marrow in our previous work [Filina et al., 2022, doi: 10.1016/j.cellsig.2021.110205]. Increased levels of phosphorylated isoforms of these enzymes were shown in response to fMLF in low concentration activating Fpr1. Kinetics of MAPKs activation in Fpr1 signaling had similarity to the curve of ROS generation. Also, it was revealed that MAPKs inhibitors (SP600125, FR180204 and SB203580 which we used in the presented work) suppressed significantly NADPH-oxidase related ROS generation initiated by fMLF. These facts indicate: (1) activation of JNK (p38 and ERK) by Fpr1 in neutrophils and (2) role of JNK (p38 and ERK) in activation of NADPH- oxidase in neutrophils. The evidence was given in the literature: coupling of FPR with three cascades of MAPK signaling in activation of NADPH oxidase has been shown in HL-60 cells differentiated to neutrophils [Rane et al., 1997]; other authors observed that fMLF (0.1 µM, preferential activation of FPR1) increased the level of p-JNK in human neutrophils and JNK inhibitor JNK-IN-8 suppressed superoxide generation [Lai et al., 2021, doi: 10.1016/j.jep.2020.113224]. Also in other cells, e.g. podocytes, over-expression of FPR1 aggravated high glucose-induced up-regulation of p-p38, p-ERK and p-JNK [Zhang et al., 2022, doi: 10.1177/15353702211047451] indicating coupling of Fpr1 and MAPKs.

SP600125 as JNK inhibitor was reported by Bennett BL et al. [Bennett et al., 2001, doi: 10.1073/pnas.251194298]. We used it as JNK inhibitor in neutrophil ROS assay in our previous work [Filina et al., 2022, doi: 10.1016/j.cellsig.2021.110205]. Also other authors used SP600125 to inhibit JNK: SP600125 (SB202190) prevented JNK (p38MAPK) phosphorylation in mouse bone-marrow- derived macrophages treated with LPS [Xian et al., 2021, doi: 10.1016/j.immuni.2021.05.004]. SP600125 suppressed significantly lipophosphoglycan induced JNK phosphorylation and decreased ROS production in neutrophils from mouse bone marrow as well as inhibitor of NADPH-oxidase (DPI) [Zhang et al., 2021, doi: 10.3389/fmicb.2021.713531]. The inhibitor decreased JNK phosphorylation in transformed cell lines [e.g., Kim et al., 2012, doi: 10.1371/journal.pone.0046208].

FR180204 as ERK1/2 inhibitor was reported by Ohori et. al., and used by Filina et al. [Ohori et. al., 2005, doi: 10.1016/j.bbrc.2005.08.082; Filina et al., 2022, doi: 10.1016/j.cellsig.2021.110205]. Authors demonstrated that FR180204 was an ERK-selective and cell-permeable inhibitor, and could be useful for elucidating the roles of ERK as well as for drug development. [Ohori et. al., 2005]. To confirm the role of MAPKs in NADPH activation and development of respiratory burst, other authors studied the effect of specific inhibitors of MAPKs (SB203580, FR180204 or SP600125, which are the selective and potent inhibitors of p38a/b, ERK1/2 and JNK, respectively) on ROS production stimulated via Fpr1 or Fpr2 and collated with their role in PMA-induced respiratory burst. All used inhibitors suppressed Fpr1-mediated respiratory burst in concentration-dependent manner with rather low IC50 that indicated participation of inhibited kinases in upregulation of NADPH oxidase activity [Filina, Y. et al., 2022, doi: 10.1016/j.cellsig.2021.110205].

Point 4: If it’s first time using SP600125 in luminescence-based ROS assay in neutrophils, additional experiments should be provided, including:

1) Direct antiradical effect of SP600125 in cell-free system (enzymatic and non-enzymatic system of ROS generation).

2) Effect of SP600125 on ROS production in neutrophils, activated via different pathways (receptors and non-receptor).

3) Test other JNK inhibitors with different scaffold on Fpr1 in neutrophils.

4) Test the inhibitor in cell-based non-luminescent assay.

Same experiments should be provided for other used inhibitors.

Response 4: We understand that inhibitors in ROS testing must have certain properties and not have others. Extensive pharmacological analysis is demanded to characterized specificity at obvious activity in enzyme and cell-based assays, also absence of activity as ROS scavenger or enhancer, lack of interaction with ROS probe or inhibition of upstream pathways. This is partially provided by creators and manufacturers. It is not first time using SP600125 and other inhibitors in chemiluminescence-based ROS assay in neutrophils in our researches [please, see, e.g. Filina et al., 2022; Filina, Tikhonova et al., 2022]. Usually, when starting to work with one or another inhibitor, we analyze the literature with its use, manufacturers' websites and conduct tests for a direct anti(pro)radical effect in a cell-free assays (Fenton reaction and xanthine / xanthine oxidase reaction). If an inhibitor in a cell-free environment affected the intensity of ROS generation in one direction or the other, we do not use it. Earlier we ascertained that used inhibitors themselves did not affect significantly spontaneous level of ROS in the cells and the ROS production in cell-free samples. We do not include these results in manuscripts since it is out of the scope of our investigations, but is one of the laboratory practice.

Point 5: Indeed, a conclusion, based on “specificity” of a particular inhibitor could be overestimated. For example, SP600125 could inhibit other kinases.

Response 5: Really, specificity of SP600125 to JNK is discussed in literature. Bennett and colleagues analyzed biochemical profile exhibited by the agent in cell-based assays. They demonstrated that SP600125 blocks JNK and not other inflammatory-signaling cascades, and seems to have a selectivity of at least 10-fold for the JNK pathway [Bennett et al., 2001]. Wu et al. noted that some ATP-competitive inhibitors of JNK, such as SP600125 and AS601245, are widely used in vitro; however, this type of inhibitor lacks specificity as they indiscriminately inhibit phosphorylation of all JNK substrates [Wu et al., 2020]. Nevertheless, the latter conclusion does not contradict the first. We changed related fragment in the “Discussion” section.

Point 6: Authors discuss (see Lines 363-366) that increased ROS generation could be caused by hyperglycemia. However, it is not at all clear how high blood glucose can affect ROS production in isolated neutrophils.

Response 6: Previously, we and other authors demonstrated that chronic hyperglycemia led to increased ROS production and modified various neutrophil functions both in whole blood and isolated neutrophils [Tikhonova et al., 2020, doi: 10.1016/j.freeradbiomed.2020.06.014; 2022, doi: 10.1016/j.freeradbiomed.2022.09.031; Ferreira et al., 2012, doi: 10.1002/cbf.2840; Mikhalchik et al., 2021, doi: 10.1007/s10517-021-05147-x] .As the cells are exposed to high glucose for a long time in the blood of patients with diabetes mellitus and animal models of disease, their functioning, ROS generation including, are changed not only in the whole blood but after isolation, in particular as a consequence of irreversible protein glycation [Osawa and Kato, 2005, doi: 10.1196/annals.1333.050; De Toni et al., 1997, doi: 10.1111/j.1749-6632.1997.tb46264.x]. It is known that receptors for advanced glycation end-products mediate the generation of ROS via activation of NADPH-oxidase. leading to priming neutrophils.

Point 7: Authors discussed about different number of Fpr2 on cell membrane, but didn’t test this. Indeed, a direct determination of number of Fpr1/Fpr2 receptors on neutrophil surface in ORM in comparison with control mice should be first experiment for this study.

Response 7: Thank you for this valuable comment. Starting the study, we did not know whether we would obtain differences between the controls and ORM group in our experiments. Fortunately, the differences has been discovered. It is logical further to determine the expression levels of both FPR1 and FPR2 genes, and proteins in controls and ORM, as well as the expression levels of NADPH-oxidase subunits and signaling participants, should be determined together with their (co)localization. We added it to the the “Conclusions” section as limitations and perspective directions for further research.

Point 8: Please, use abbreviation ROS instead RS.

Response 8: We did it.

Point 9: Typo: “PLC, PLC…”  (Line 264)..

Response 9: We corrected it.

Round 2

Reviewer 3 Report

I thank the authors for their answers.

I recommend that the authors next time use an alternative way to activate NADPH oxidase (eg PMA to activate protein kinase C). Indeed, if chronic hyperglycemia leads to an increase in ROS production and modifies various neutrophil functions in isolated neutrophils, then this effect may be independent of FPR. I would like to get an answer from the authors, why do they think that FPR, and not other pathways of NADPH oxidase activation, are important for the pathology under study?

Author Response

Point 1: I thank the authors for their answers.

I recommend that the authors next time use an alternative way to activate NADPH oxidase (eg PMA to activate protein kinase C). Indeed, if chronic hyperglycemia leads to an increase in ROS production and modifies various neutrophil functions in isolated neutrophils, then this effect may be independent of FPR.

I would like to get an answer from the authors, why do they think that FPR, and not other pathways of NADPH oxidase activation, are important for the pathology under study?.

Response 1: Indeed, in our study FPRs have become the main focus for the following reasons. Among numerous neutrophil receptors (GPCR, Fc-receptors, adhesion receptors, cytokine receptors, innate immunity receptors) only GPCRs (receptors of formyl peptides, LTB4, PAF, C5a, IL-8) activate the generation of ROS [Futosi et al., 2013]. Also, in neutrophils adrenergic, opioid, insulin, nicotine, and muscarine acetylcholine receptors (modifying, not activating respiratory response) can be listed. In the hierarchy of these receptors to activate NADPH-oxidase, FPRs prove to be the most powerful. Phosphorylation is a basic mechanism of activation and regulation of NADPH-oxidase activity. All subunits of NADPH-oxidase can be phosphorylated, and PKC plays a main role. Besides, phosphatidic acid-activated kinase, p21-activated kinase, MAPKs, PI3K/PKB, tyrosine protein kinases can also participate in the activation/regulation of NADPH-oxidase activity, especially under pathological states. Well-known pathway of NADPH-oxidase activation via FPRs includes (shortly): dissociation of Gi/0, activation of PLCβ, generation of IP3 and Ca2+ release, production of DAG (PKC activation). Other mentioned signaling components can be also involved. The comparison of FPR1 and FPR2 signaling to NADPH-oxidase has shown the involvement of different participators (e.g. regulating cytoskeleton). We did not find the works concerning signaling (via FPRs or other receptors) to NADPH-oxidase in ORM neutrophils, the focus is on “insulin resistance”. However, FPRs were studied in the obese state, what we cited in the Introduction: “FPR1 gene was downregulated in obese human, besides CYBB gene coding NADPH oxidase membrane subunit gp91phox was also downregulated in obesity [12]”. FPR2 was mentioned also (for macrophages, not for NADPH oxidase). Levels of microbiota-produced fMLF, an agonist of FPRs with multiple targets in an organism, were increased in the blood and intestine in mice fed a high-fat diet [Wollam et al., 2019, doi: 10.2337/db18-1307]. It indicates FPRs relevance in high-fat diet eating. Therefore, first of all, we concentrated on this type of receptors. We studied four groups of animals: controls and controls with acute inflammation, ORM and ORM with acute inflammation (that exacerbates the pathological process), and expected that FPRs, being involved in the pathogenesis of the various inflammatory conditions presented in our work, and having branched signaling to NADPH-oxidase, would allow us to distinguish obesity-resistant state from controls. Certainly, it is interesting to investigate other signaling ways of NADPH-oxidase activation in obesity resistance that will be the scope of our further studies.

Round 3

Reviewer 3 Report

The paper could be accepted in the present form.